# Brothels as Sites of Third-Party Exploitation? Decriminalisation and Sex Workers' Employment Rights

**Gillian Abel ***[ID] **and Melissa Ludeke**

Department of Population Health, University of Otago, Christchurch 8140, New Zealand; Melissaludeke@gmail.com
* Correspondence: gillian.abel@otago.ac.nz

**Abstract:** Decriminalisation is arguably essential to protecting the human rights of sex workers. Nonetheless, there are suggestions that decriminalisation has less influence on sex workers' experiences of working than many assume. This paper explores management practices in brothels in the context of decriminalisation in New Zealand, focusing on sex workers' employment status, managerial control and agency. We interviewed 14 brothel operators and 17 brothel-based sex workers in this study. The findings suggest that there remain challenges for sex workers in that brothel operators treated them as employees rather than independent contractors. Brothel operators retained control over shift times and pricing of services, and working conditions were unclear. Most sex workers understood their rights, but when operators impinged on their rights, it was often more expedient to move place of work than make an official complaint. However, decriminalisation did have a meaningful impact on the way sex workers negotiated potentially exploitative dimensions of brothel-based work. Decriminalisation has provided the context where it is possible for sex workers to experience safer and more supportive work environments than they otherwise might, where they can (and sometimes do) contest managerial control.

**Keywords:** decriminalisation; employment; human rights; sex work; exploitation

## 1. Introduction

Amnesty International formally adopted a policy in 2016 endorsing the full decriminalisation of consensual commercial sex to protect the human rights of sex workers globally. New Zealand is the only nation that has decriminalised sex work (although one state and one territory in Australia have done the same). Several countries have legalised sex work, but there are important differences between decriminalisation and legalisation. Decriminalisation applies to sex workers in all sectors of the industry, whether this be brothel-based, street-based or online-based sex workers. Third parties, including clients, brothel operators and pimps are also decriminalised. Legalisation, however, is only relevant to licensed brothels. All sex workers and third parties outside of licensed brothels remain criminalised. Some countries adopt the so-called "Nordic Model", which decriminalises sex workers but criminalises all third parties.

The literature to date on the different ways of legislating sex work and their pros and cons is prolific (e.g., Östergren 2017; Pitcher and Wijers 2014; Sanders et al. 2018; Scoular 2015; Sullivan 2010; Benoit et al. 2019). Agustin questioned the centrality of legal and regulatory approaches when thinking about sex workers' agency and rights in her blog, seeing it as "bizarrely irrelevant, except for its symbolic value . . . prostitution law is often vague and unenforceable, in the end having less impact than people assume" (Agustin 2009). Scoular (2010, p. 38), however, whilst agreeing with Agustin that law should be de-centred, states that it cannot be excluded as "it is imbricated in the everyday world" and "operates alongside other normative ordering practices to shape subjects, identities, practices, and spaces". The ongoing debate that has ensued has resulted in calls for research

to consider nuanced experiences and environments that shape sex workers' agency in the labour market, and the broader cultural landscapes and spaces within which sex workers operate (Hammond and Attwood 2014; Bungay et al. 2011; Agustín 2005; Weitzer 2012; Van Meir 2017).

Most sex workers work in the indoor sector (Bowen 2015; O'Doherty 2011; Bungay et al. 2011; Harcourt and Donovan 2005). Safety is an important "pull" factor for indoor work, particularly for those new to sex work (Brents and Hausbeck 2005; Zangger 2015). Brothel-based sex workers experience little to no violence, especially when compared to their street-based counterparts (Prior et al. 2013; O'Doherty 2011; Brents and Hausbeck 2005; Church et al. 2001; Sanders and Campbell 2007; Zangger 2015; Prior and Crofts 2014). Brothels provide a context of protection through "collective control" (Sanders and Campbell 2007, p. 10), where the physical environment and material resources affords a sense of security and confidence. This includes safety measures such as presence of other staff, security cameras, alarms and opportunities to screen clients thoroughly (Zangger 2015; Brents and Hausbeck 2005; Sanders and Campbell 2007; Sullivan 2010; Van Meir 2017). Nevertheless, brothels also hold the greatest environmental potential for managerial control over sex work labour. Largely, the ways in which sex work is managed affect the safety and protection afforded to sex workers (Sanders and Campbell 2007; Orchiston 2016; Bungay et al. 2011; Brents and Hausbeck 2005; Bruckert and Law 2013).

Before decriminalisation in New Zealand, brothel-based sex workers (or parlour-based as it was known then) were subject to particularly "manipulative management strategies . . . (which pushed) them towards risky practices" (Plumridge 2001, p. 212). Sex workers' agency was severely restricted. The Prostitution Reform Act (PRA) (2003), which decriminalised sex work, intended to safeguard sex workers' rights by limiting the power of brothel operators. The Act provides protections for sex workers by making inducing or compelling persons to provide sexual services an offence (Section 16) and stating that a person may refuse to provide a sexual service at any time, even if they had previously entered into a contract to provide such service (Section 17). Research carried out shortly after decriminalisation, however, suggested that decriminalisation had not eradicated poor management practices and that there was inadequate monitoring or enforcement of provisions under the PRA (Mossman 2010). Sex workers did, however, exercise some agency, as brothel operators in Mossman's study reported that they found "controlling" sex workers in the new context difficult. There were tensions about how they could run a viable business when sex workers had the right to refuse to provide certain services or refuse to see a client at any time (Mossman 2010).

A more recent study in New Zealand (Zangger 2015) explored labour conditions for sex workers in Auckland in both the managed and private sector. Brothel workers in this study expressed concerns about three key areas of poor practice. Firstly, they indicated that their right of refusal was subject to limitation and "good reason". Secondly, managers failed to uphold shared responsibility for safe sex practice. Thirdly, there was coercion to work longer and less flexible hours and/or with more clients than desired through the utilisation of shift fees, fines and bonds (including penalties for arriving late or leaving early). Despite this, sex workers in that study considered brothels a favourable industry entry point. Inexperienced workers preferred to work under management rather than privately, as working privately involved greater responsibility and additional costs. This is consistent with the findings of a study of 772 sex workers in New Zealand, where brothel-based sex workers saw the benefits of working in a brothel as outweighing the trade-offs (Abel and Fitzgerald 2012).

A review by Harcourt et al. (2005) on different legislative approaches across Australia concluded that while not a perfect approach, the decriminalised context in New South Wales appeared to have the fewest pitfalls and was the most effective in terms of improving health outcomes. More recent research by Orchiston (2016) and Sullivan (2010) has compared the impact of decriminalisation and legalisation in different states in Australia (New South Wales and Queensland). Orchiston concludes that ultimately, neither legalisation nor



decriminalisation have "any substantive connection to improved labour rights for sex workers" (p. 2) due to the lack of regulatory enforcement of labour protections. Furthermore, Orchiston argues that brothel operators "use the cloak of legality to operate openly and claim legitimacy, without having to adhere to the same requirements as other businesses" (p. 11), while the vague status of workers as employees or independent workers bears considerably on their access to legal protection (Murray 2001). Sullivan notes that while both legalisation and decriminalisation have contributed to an economic integration of the industry, persistent "social ambivalence" (p. 103) and stigma prevent the recognition of sex work as legitimate "work" under either framework.

Overall, available research indicates that while decriminalising brothel-based sex work can increase the safety of workers (Abel 2010), alone it is not sufficient to improve working conditions (Mossman 2010), and intentions can be lost in the implementation of legislation and regulation (Scoular 2010; Wagenaar and Altink 2012). There remains a limited understanding of the power dynamics operating within the indoor sector of the sex industry (Harcourt and Donovan 2005; Weitzer 2009). Few studies have explored the management, organisation structures and working conditions of brothels (Weitzer 2009; Cruz 2013), particularly in decriminalised contexts (Cruz 2013). New Zealand provides an opportunity to explore dimensions of brothel-based sex work that have changed or persisted now that decriminalisation is well-embedded. This paper focuses on sex workers' employment status and agency within brothels in New Zealand.

## 2. Methods

The data for this paper draw from a study aimed at exploring employment rights of brothel-based sex workers in New Zealand's decriminalised sex work environment. The University of Otago Human Ethics Committee granted ethics approval (Ref no: 15/169). A community-based participatory research approach (CBPR) was taken to the research. CBPR bears many similarities to other participatory approaches in that it is participatory, empowering and achieves a balance between research and action. However, a community-based organisation who represent the larger community are involved in the development of research questions and the research design, rather than the participants themselves. All forms of participatory research are acknowledged as best practice, and most ethically appropriate, when carrying out research in the sex industry (Benoit et al. 2005; Wahab and Sloan 2004; O'Neill and Pitcher 2010). New Zealand Prostitutes' Collective (NZPC) was our community partner. They were partners in identifying the research questions, had input into the design of the data collection tools and the recruitment of participants. Participants are less likely to be exploited when there is power-sharing in a partnership-based research relationship and the findings of the research are more likely to reflect their perspectives (Liamputtong 2007).

This was a qualitative study using in-depth interviews to capture the perspectives of 17 sex workers on working as independent contractors in a decriminalised environment. The interviews were semi-structured with no specific questions asked, but topics covered included: their relationships with brothel operators/managers; contractual arrangements; experiences within brothels; and perceptions on stigma. Sixteen of the sex workers interviewed identified as female and one as non-binary gender. In addition, interviews were conducted with 14 brothel operators (eight males and six females) around topics such as: business practices; contractual arrangements; regulations relating to operating a commercial sex business in a decriminalised environment; and perceptions on stigma. Four of the female brothel operators/managers and one male operator had worked as sex workers before operating or managing a brothel; 13 had only operated a business in a decriminalised setting. The first-named author conducted the interviews in the cities of Auckland, Christchurch and Wellington between February and August 2016. Most interviews occurred in NZPC offices, but some took place in a brothel. Each participant was given NZ$40 in appreciation of their contribution to the research.

All operators and sex workers were operating/working within brothels that operated openly and most situated amidst other businesses in the city (in Auckland and Wellington) or on the edges of the city or industrial areas, particularly in Christchurch where the 2001 earthquake had forced central city brothels to move out of the central business district. All operators and sex workers had links into NZPC. None of the operators or sex workers were from the more informal, suburban areas, where migrant (mostly Asian) sex workers work, and which are often more isolated from NZPC and other service providers. The findings of this study therefore only reflect the experiences within New Zealand's "visible" brothel-based sex industry, a limitation of this research. It is of course likely that the experiences of sex workers who work from hidden brothels in the suburbs, and who may not be New Zealand citizens, will be very different and more precarious, as described elsewhere (Abel and Roguski 2018; Armstrong et al. 2020).

Interviews were audio-recorded and fully transcribed. Participants' anonymity was protected by providing them with pseudonyms and removing all identifying information from the transcripts. Thematic analysis was then undertaken, which is a method used extensively in qualitative research to identify, analyse and report patterns in data (Aronson 1994; Braun and Clark 2006). Both authors read and re-read the transcripts to familiarise themselves with the data and then coded for features of specific significance. For this paper, we were interested in the talk of both sex workers and brothel operators on management practices. Codes included (amongst many others) the law, contractors, employees, shift fees and fines, resistance, co-dependency and reporting. Codes were then collapsed into overarching themes. This paper focuses on three themes: employment status, the economics of brothel-based work and agency in a brothel setting.

## 3. Results

### 3.1. Employment Status

Private homes are increasingly becoming the most popular working environment for sex workers in many quasi legal or decriminalised settings, and this shift has been documented in New Zealand since decriminalisation (Abel and Fitzgerald 2012; Prostitution Law Review Committee 2008). Sex workers value working independently as it increases control and freedom, financial independence, flexibility over commercial working spaces, and they can operate discretely (Hubbard and Prior 2012; Prior and Crofts 2014). Many indoor workers aspire and work towards being able to work independently (Bernstein 2007a, 2007b). Nonetheless, the sex workers interviewed in this study saw working in brothels as a way of "out-sourcing" tasks, such as advertising and making bookings, activities that they did not necessarily want to perform themselves. It also provided a place to work away from their own home:

> I don't want to be managing my ads. I don't want to be answering the phone when I'm at home and make the arrangements and find some premises to see the clients at. It's too much hassle, so I see the parlour getting a cut as kind of out-sourcing. (Milly, sex worker)

Nevertheless, many of the sex workers in this study were concerned that brothel operators and managers tried to exert too much control over them, which was not in line with their independent contractor status. Sex workers in New Zealand have always been independent contractors. This is consistent with the norm in other countries, regardless of the regulatory framework (Cruz 2013; O'Connell Davidson 2014; Cruz et al. 2017). Brothel operators positioned sex workers as no different from any other contractor:

> No different than if you'd employed a plumber to come and fix your plumbing. They're an independent contractor. You don't ask them to sign a contract or anything to come and do that. They look after their own finances. They give you a bill and you pay it. That's how we work it. (Harry, operator)

Perhaps the comparison to contracting a plumber is erroneous. A plumber is engaged to fix a problem and then goes away, only returning when there is another problem. Brothel

operators engage sex workers to provide a service on a regular basis to others accessing the brothel. Brothel operators are able to maintain a high level of managerial control and take limited responsibility for sex workers through the "loophole" of independent contracting (Murray 2001; Sullivan 2010; Cruz 2013; Cruz et al. 2017; Bouclin 2004; O'Connell Davidson 2014; Pitcher 2015). The setup has been termed "notional" self-employment (Sanders and Campbell 2007), "disguised" employment (Orchiston 2016), "false" subcontracting and "dependent" self-employment (Sullivan 2010) and is considered to leave sex workers highly vulnerable to coercive management practices (Murray 2001; Orchiston 2016). It is common that managerial control is exercised over the hours worked (Orchiston 2016; Sullivan 2010). Some sex workers in this study worked in "appointment only" businesses and so only went into the brothel when they had appointments with clients, as one brothel operator explained:

> *We do it all by appointment only, so they don't have to do shift work. They pick their own hours, so they just choose. Like today, Monday, tomorrow they'll be texting, "Hey, [Ingrid], my hours this week are this, this and this. Like I'm free all day Wednesday except I've got a lecture between 2 and 3", so I work around that.* (Ingrid, brothel operator)

Sex workers in "appointment only" businesses seemed to be happy with this arrangement. Those who worked in "walk-in" brothels, however, had to agree to pre-determined shifts. Brothel operators in walk-in brothels suggested that they were mindful of the requirements of the PRA and so left it up to individual workers to tell them when they would be available:

> *The Prostitution Act specifically says these girls, they can come and they can go anytime they want. So I just follow actually the rules to tell the girls, "I don't actually ask you to be here all the time, but at least if you guys can give me the timetable for the week, so it's easier for us to manage your bookings. So you save your time, and we save our time".* (Robert, brothel operator)

However, many sex workers in walk-in premises contradicted this. They indicated that management tried to exert control over them by requiring them to work ten-hour shifts. Furthermore, they claimed that brothel operators used unfair rostering and did not allow them to take leave. They argued that this challenged their independent contractor status:

> *They've got a lot to answer for, certainly they do. When we're not employees in a brothel and yet they're treated like employees and without any of the rights of an employee . . . expected to work ridiculous shift hours. Like most, every brothel I know has a minimum requirement of 10 h which is ludicrous. . . . . Some girls would be a lot like, "Oh I want some time off and I'm afraid to ask". They'd say they wanted Friday night off and he'd say, "No, I'm doing the roster. No, you can't have Friday off, you can have Saturday off". So yeah, they would, they would say no sometimes to girls, which if you're a contractor, they haven't got the right to do.* (Cynthia, sex worker)

Most contractors are able to take a meal break during a ten-hour shift. However, Nyla received a telling off when she went out to collect some lunch:

> *The receptionist, when I came back, said, "Oh where were you?" and I said, "Oh, I went to go get some food", and she was like, "You can't leave. You know, you can't just leave". And I was like, "Oh I was just getting some lunch", and she was like, "No, you know, we can't have the girls leaving, blah blah blah" . . . . She kind of tried to tell me off. I didn't want to have an argument with her cause I wanted to work there. So I was just like, "Okay, you know, cool, I'll just bring my lunch next time, whatever". I didn't really care about it, but I was like, "That's not cool. People should be allowed to leave and get their lunch. And you wouldn't have that in any other workplace".* (Nyla, sex worker)

There was a quandary for brothel operators who argued that they were trying to run a viable business that required sex workers to be on the premises should clients arrive, but they were aware that under the PRA, they could not compel anybody to be there to provide

a sexual service. While they may negotiate with sex workers to be at work for an entire 10-h shift, there was little that they could do to enforce this. Written contracts were rare and seen as unenforceable. Some sex workers in this study, like Eve, said they had signed one but could not recall what was in the contract and did not have a copy of it themselves:

> *They kept the contract and I never received a copy, and in the last 6 weeks I tried to follow up and obtain a copy of that contract, which I never ended up getting . . . to be honest I don't think I looked through the contract as such very well.*  (Eve, sex worker)

One brothel operator questioned the value of the contract anyway given that some workers signed them with false names:

> *I mean it's a waste of paper really [contracts]. We were hiring women, they were coming and going all the time, and they only signed it by their working name anyway, so what is that? They could say, "That's not me. I'm not Christina. No, my name is whatever". So what's the point, you know. I mean, yeah.*  (Ingrid, brothel operator)

Not all sex workers wanted written contracts as they saw it as a threat to anonymity. Nyla, in particular, indicated that she claimed a benefit in addition to working in a brothel and was concerned that government officials would be able to access information from brothels on contracted workers:

> *One of the sex workers had thought that contracts were a good idea for workers' rights, so she was sort of into union-y stuff and she said, "We should have contracts blah blah blah" . . . I was like, "Yeah, and what about if, I don't know, Ministry of Social Development come in and say, 'We want to see your records', and then my name's on file or record and I'm on a benefit". I wasn't going to sign. I was like, "I do not want to put my name on that, and you know, I'm not really keen to work under those sort of conditions".* (Nyla, sex worker)

House rules often took the place of contracts and were mostly about not using drugs on the premises and always using a condom for penetrative (and sometimes oral) sex. Operators indicated that these rules were normal and applicable to any other workplace, and if broken, they were within their rights to tell the sex worker to leave:

> *Everything that is in a normal workplace is in our place. You come to work stoned, you get sent home. You do drugs at work, you're fired immediately. You get verbal warnings, just like anywhere else for things. But 'cos they're an independent contractor, of course we can get rid of them immediately. Okay, but at the same case, they can leave immediately. There's no forcing of anybody to be anywhere or do anything.*  (Harry, brothel operator)

There remain concerns about the employment status of brothel-based sex workers in New Zealand. Some participants saw the requirement to work long shifts as a threat to their independent contractor status. However, there was ambiguity around contractual arrangements. Whilst verbal contracts may be legally binding, enforcement is difficult as the terms, conditions and rights of the contractor are not clear (Prostitution Law Review Committee 2008). It seems, therefore, that decriminalisation has not been successful in advancing a formalisation of the working relationship between sex workers and operators. The control management exercised over sex workers' income was one particularly grey area in the relationship between the two, and we discuss this in the following section.

### 3.2. The Economics of Brothel-Based Work

Most independent contractors are able to negotiate a price for their services with business owners. Business owners then add a percentage on to this price, which would be their cut. However, sex workers have no say in setting the price of particular services in brothels. The brothel operator sets the price as well as the house cut, which is standard for all sex workers working on the premises. Operators in this study took cuts ranging from 30% to 40%. They justified the size of the cut as necessary because of taxation and overhead costs:

*We supply everything. And then you get the girls that don't appreciate anything . . . they don't see the overheads, the advertising, you know, and I have to pay some tax.* (Frank, operator)

*We charge the girls [a] service fee, like providing them the premises, safety, guardians, and actually we put all the advertising fees on us, so all these costs altogether, we charge the girls $69 for each job for an hour.* (Robert, operator)

Managers and operators commonly explained their working arrangement as sex workers paying them for a service as opposed to the other way around:

*Obviously we run the place more like a hotel, more like a motel. I hire out the rooms . . . . They're free to just make a booking and come when they have a booking. We have a little lounge where they can wait if they wish. . . . They pay us for the room.* (Burt, operator)

Clients pay their money to reception when they arrive and sex workers usually receive their money, minus the cut, at the end of the evening or the next day, depending on the arrangement with the establishment. Some sex workers do experience some problems getting their money on time. "Safe malfunctions" and other management lapses led to Bridget feeling devalued:

*There's supposed to be a window when you collect your pay. It's supposed to be between 8 and 9 [the next morning], although if you're working that night, you can just go in later in the night. But the safe malfunctioned and my pay was locked in, and although I have enough in savings, I didn't desperately need that money right then, I was still really upset by this just because it had come after a booking that had been allowed to go overtime, and just, and come after two weeks of still no new light bulb and just other things where it made me feel really not valued as a worker. And it stresses me out not having access to my money, I guess, in that I've had times when I've been quite broke doing this work, and so I don't like management holding on to my money when it's mine.* (Bridget, sex worker)

In addition, some brothels took a further cut when clients used a credit card and sometimes delayed payment to sex workers. This was to cover the bank fees for credit card charges and to allow time for the transaction to clear:

*If someone pays card it's $30 extra, but it doesn't, I don't think it costs that much to do a card transaction.* (Carrie, sex worker)

*Apparently credit cards can take quite a few business days for it to process through and the funds to become available to the owners, it can be anywhere from 3 to 5 business days before the girls will receive their money. Now as far as I'm aware, legally they're not actually allowed to hold on to money like that, but there are places that do do that and the girls just go along with it.* (Roz, sex worker)

Costs for advertising are supposed to come from operators' cut of sex workers' money. Brothel-based sex workers are dependent on operators for advertising their services and securing business for them. Most operators spent large sums of money annually on advertising, and sex workers were reasonably happy with this. However, a few sex workers were frustrated at having to take control of advertising themselves. Pam considered that management did little to attract clients. They did not have a company website and did not advertise individual workers on the main sex work platforms. Consequently, the number of clients had diminished considerably, with the knock-on effect of decreased income. She argued that in order to make a living, they had to pay for their own advertising over and above the standard cut:

*I don't think they're doing much to generate more business for the girls. You know, they're on and on and on about, you know, having staff, but, you know, you need to spend money in the business to make money as well . . . the girls are having to do it, they're having to do it for themselves to generate the business.* (Pam, sex worker)

Management are also able to make price changes with no prior consultation. One sex worker said that overnight her pay per hour appointment went down by $15:

> *[I was paid] $135 an hour, and then I guess something changed, I don't know, whether their [management's] rent went up . . . And then [the manager] got me at the beginning of one of my shifts and goes, "Now look, we've had management change. Do you still want to work here because now the hourly rate you'll get paid is $120"? Yeah, so it could be what lots of companies do now, just cost cutting and trying to keep some more money in their pocket.* (Vicky, sex worker)

Brothel operators and managers have traditionally applied economic sanctioning as a mechanism to assert power and control over sex workers (Cruz et al. 2017; Orchiston 2016; Sullivan 2010; Dziuban 2016). Orchiston (2016) argues that this control is financially rewarding for operators, shifting economic risk onto sex workers in order to reduce overheads and maximise profits. This was also the case in New Zealand prior to 2003 (Mossman 2010; Prostitution Law Review Committee 2008; Plumridge 2001). The PRA now explicitly prohibits any form of coercion to provide services. Paul was the only operator who admitted to fining sex workers:

> *Look, it's a hard, hard industry. Yes, we fine them, because we're still a business, you know. If I don't make $50,000+ a month, I'm losing money, you know, and you have to have girls on the floor for that. So that's why parlour owners fine.* (Paul, brothel operator)

One sex worker said that an operator fined if you repeatedly did not arrive for shifts "because he has to pay for the ads. And so he'd fine you like the amount of money he'd put into an ad if you didn't show" (Deena). Another sex worker said that the receptionist at the brothel she worked at had fined her for being late for a shift, but knowing her rights, she complained to management who immediately reimbursed her:

> *It was just a receptionist that did the fine to kind of spite me because she thought I was rude to her on the phone when she called me asking me where I was. So she just did it because she was angry, and then when I talked to them—I talked to her about it and she just wasn't really listening to anything I had to say. So I was like, "Well fine, I'll talk to the manager". So I talked to the manager and she was like, "Oh I'm sorry. I'm giving you your money back straight away".* (Serena, sex worker)

Although sex workers knew they had rights and did not have to put up with fining, they had to think of possible repercussions if they did try to recoup their losses. As independent contractors, they are required to declare their earnings and pay taxes but many do not:

> *I was losing a lot of money . . . getting too much money taken off me. I wanted to try and get it back, but, you know, I was scared, I guess, that if I did something about it, then they might report me to IRD, you know, for not—cause as an independent contractor, you know, I was not paying taxes on the money that I earned. Even because I had a child as well, you know, and if I got exposed for working in the sex industry, you know, it would look bad.* (Eve, sex worker)

There is, thus, a great deal of management control over sex workers' earnings, and some practices continue to infringe on sex workers' rights. It is widely argued that decriminalisation protects sex workers' rights and enables them to organise for better working conditions (Abel 2010; Baratosy and Wendt 2017; Kim and Alliance 2015; Wijers and van Doorninck 2016). In many ways, sex workers in this study displayed a level of control and autonomy over their work, yet there were also times when there they were hesitant about exercising their rights.

*3.3. Rights in the Workplace*

All of the operators interviewed in this study argued that they were "good" and ethical business owners and treated sex workers fairly. All were well aware of the requirements for operating a brothel and the inbuilt protections for sex workers under the PRA. They indicated that sex workers who worked in their brothels understood their rights and felt that with NZPC's support, they knew how to enforce those rights. Frances argued that

there would always be "dodgy parlours", but NZPC mitigated this by providing support and information on rights to sex workers:

> *. . . in saying that, like the girls . . . once they've been around for a little while, they do know their rights. . . . . I send my girls in here (NZPC), . . . as long as PC's [NZPC] around, the girls will know that they can't be treated like that.* (Frances, operator)

Although some sex workers in this study indicated that there were still some exploitative practices happening within brothels, overall, they felt that management was good and in most cases, respectful:

> *I feel like actually of those 6 owner/manager people [that she had worked for], 5 of them have actually been quite, quite fine and respectful. Yeah, actually pretty cool towards the sex workers, even though, you know, some of the places I haven't loved working at, but I feel like that actually the owners have been, besides that one, have all been really good.* (Nyla, sex worker)

Sex workers expected good treatment from operators, as they had the flexibility to leave at any time:

> *Being self-employed and choosing where we want to go, there's got to be something there to keep us, keep us working there, otherwise we're like, "Oh I'm just going to give this other place a call and see if they can take me tonight".* (Milly, sex worker)

Sex workers commonly move between venues to access improved working conditions and environments (Bungay et al. 2011). Managers and operators in this study found depending on a highly mobile workforce challenging. This is consistent with New Zealand research carried out shortly after decriminalisation in New Zealand (Mossman 2010). They recognised that treating sex workers well was a key business strategy to retain as well as attract workers:

> *We recognise the fact that without the girls, we don't have a business. So if you're going to treat them like shit and they all leave, then it's your own fault . . . If you treat them well, and they all stay, then kudos to you . . . I think the greatest thing that's come about from all this [decriminalisation] then must obviously be the fact that the girls do have a choice . . . they are free to make that choice, and if you run your business poorly, one would think it's going to hit the wall because you're not going to have workers.* (Burt, operator)

Although managers have traditionally exercised high levels of control over the "aesthetic labour" of sex workers through explicit rules on appearance (Hardy 2013, p. 47), the workers in this study indicated that while management could offer suggestions, they held the ultimate control over their self-presentation:

> *They definitely would give suggestions for other things that we should be wearing and stuff like that. But with us it's, I mean I normally wear sort of like an evening dress or stuff like that, so we can choose. We don't have to wear lingerie or stuff like that. We can if we want, but also they don't [push].* (Carrie, sex worker)

Operators frequently backed down when sex workers vocally resisted infringements of their rights. The sex workers in this study understood that any contractual arrangement they had with management was ultimately not legally enforceable. They demonstrated an awareness of their rights and argued that sex workers less aware of their rights under the PRA were more vulnerable to be treated as employees:

> *I said I was taking all of January off because I was spending it with my daughter and going to be touring round. They said, "Then how can you do that? Have you asked for the time off"? I said, "I don't need to ask for time off, I'm not an employee, I'm a self-contractor". But see, I'm an educated woman, so I understand the differences, and there's a lot of girls that go in there, they don't have any idea what they're entitled to, what their rights are.* (Cynthia, sex worker)

Section 16 of the PRA states that no person may induce or compel anyone to provide commercial sexual services. A conviction under this section could attract up to 14 years

imprisonment. One sex worker used this section to counter an operator's insistence that they come into work when they are not well:

> I said, "Oh actually I'm not really feeling very well, so I'm not going to come in today ... " "if you don't, yeah, so then you'll have to reimburse us our share of the cost" ... That's clearly trying to pressure me into having sex when I didn't want to ... I left and I texted her and I said, "You should try running your business within the law because it's illegal for you to try and pressure me into work when I don't want to". (Nyla, sex worker)

Plumridge (2001) noted that brothel operators in pre-decriminalised New Zealand sometimes marketed particular services without consulting sex workers, which meant that sex workers had to engage in certain sexual practices against their wishes. This led to them feeling degraded. Sex workers could also not refuse to see a client—Plumridge (2001) cites one sex worker forced to accept a client who had just vomited on her. Brothel operators most frequently sided with clients against the sex worker. The operators in this study emphasised that, consistent with the intention of the PRA, sex workers now had the ability to "call the shots" in terms of what services they would or would not provide:

> I mean one of the reasons why the girls like working for us is they can work as hard as they want or as little as they want. . . . . We're now in a normal working environment. They can, at my place they work whatever hours they decide. They take breaks whenever they want. If they don't like a customer, they don't see them. There's no penalty, no nothing. There's a customer that they don't particularly like, we won't book them again for them. I mean they call the shots. They're an independent person. We can't control them in any way except for the quality of our business. The hours they work, the things—they don't have to kiss if they don't want to kiss. But that's all up to them. (Harry, operator)

> You know, they're all independent. They come in any time they want, they leave at any time they want. They do any service they like. They refuse any service they don't like, they are 100% independent. . . . . we don't force them to do anything they don't like. (Max, operator)

Operators/managers and sex workers in the study emphasised that the right to refuse any service at any time enhanced sex workers' control and autonomy in the workplace. Most operators/managers said that sex workers (and not them) controlled the services they engaged in. Brothels operated by offering a standard service package, with extra services and payment negotiated only between the worker and the client.

> All I know is now, where I work, I feel like I have total, total control. ... But no, it's never, ever, no brothel owners have ever told me to do things that I didn't want to do. (Candy, sex worker)

However, there were ways some operators used to get around "rights of refusal" whilst still operating within the law. One so-called high-end brothel-operator stated that she would not, and could not, force sex workers to do anything that they did not want to do. However, she implied that she would not have anybody working for her who was too selective about the services they would provide. This veiled threat meant that only compliant sex workers were able to work for her:

> Women who come in and I say, "Legally there is none of those things I can enforce of you. If you don't want to kiss, I cannot make you kiss a client. It's your choice. If you don't want to have a man go down on you, I legally can't make you do it, but if you choose not to do those things, you won't get work with us and this is not the agency for you". (Ingrid, brothel operator)

This operator went on to state she would not book clients for a sex worker who did not comply with her rules, saw clients privately, or generally did not fit with the "image of the agency". She would cut off their earnings to force them into moving to other premises:

> *So if for any reason that woman's not working out, her work just goes down, down and down and she leaves of her own accord. It's kind of like, "It's not really working out. Maybe you should try another agency for whatever reason".* (Ingrid, brothel operator)

Most sex workers reported being aware of options to report and lay complaints against third parties and referred to NZPC as a key source of information and support in this process.

> *From a working girl's point of view, I just think having it decriminalised has been such a breath of fresh air and such a release of stress, I guess, knowing that there are organisations like NZPC as well that have our backs, and also that we don't have to shy away from calling the police if things do get out of control with the clients.* (Roz, sex worker)

However, many sex workers remained concerned about a threat to their anonymity if they put in a formal complaint. The social stigma attached to sex work is still very much present in a decriminalised context and there remains a fear of a public disclosure of their "real name" in court proceedings:

> *I've thought about it, but, you know, too much hassle and I didn't want to get my real name involved which might have to happen. I have mentioned, you know, some small things like that happening at NZPC, but never taken it further.* (Milly, sex worker)

In many cases, NZPC act as mediators in disputes between brothel operators and sex workers to try to reach a resolution outside of the formal complaint system. Some sex workers, however, stressed the importance of standing up for their rights regardless of threats to anonymity, as it would pave the way for other workers to do so:

> *I'm a very discreet person. I have two very different lives. I have two jobs, very, very busy outside of this work as well, and the last thing that I would need is to be named and shamed or anything come out in the public eye that kind of changes people's opinion about me, especially where my other job is concerned . . . but I would definitely, I will not stand for any bullshit for myself, and when it happens to others, all I can do is encourage them . . . people forget too, you know, since they [decriminalised] this, our rights, you know, I just feel like I'm going to stomp up and down, "This is a work place, so, you know, I have more rights than you realise, buddy".* (Pam, sex worker)

One worker who did speak out and take action against a brothel manager who sexually harassed her argued that through standing up for her rights, she could set an example for other sex workers that they did not have to put up with bad treatment within the industry:

> *My thinking was that if I could show other girls that they will be listened to if they're in the same situation, like either with a boss or a client or whatever, that there is a place that will listen to them. And if it helps one girl come out and speak about bad experiences or whatever, then I'm happy . . . it just takes one person to stand up and go through it to help others come out as well.* (Kyra, sex worker)

In the main, management practices appear to have adapted to fit with the legal requirements of the PRA, but there are still questionable strategies evident in some management approaches. Sex workers, at least the ones in this study, argue that they have high levels of control through knowledge of their rights in the workplace, and there is a readiness to enforce these rights. However, they have to weigh up enforcing their rights against possible exposure.

## 4. Discussion

Brothel operators perform a necessary function for some sex workers who do not want to engage in the business-side of their work. They advertise their business as well as the individual workers within that business, which removes the onus of marketing from the sex workers. They provide a room with all the amenities so that sex workers can keep their home life separate from their working life. In other words, as Bruckert and Law (2013) have argued, sex workers are able to free up time by avoiding tasks they do not want to do.

Most sex workers in this study therefore saw their relationship with brothel operators as symbiotic.

As in most countries, brothel-based sex workers in New Zealand are independent contractors. Brothel operators function merely as the middle-person between sex workers and clients. However, sex workers in this study argued that brothel operators sometimes over-stepped this role and treated them as employees. People who are self-employed in the mainstream labour market are predominantly autonomous, but management within brothels is often to such a high level that sex workers' autonomy is questionable (Cruz et al. 2017). Means and Seiner (Means and Seiner 2015, p. 1511) argue that the only way to determine whether a worker is an independent contractor or employee is to ask the question: "How much flexibility do individuals have in determining the time, place, price, manner, and frequency of the work they perform"? Those who are able to exert control over all these variables are more likely to be functioning as independent contractors and not employees. There is no question that brothel-based sex workers in this study had some flexibility over all these variables but not total flexibility. Shift times were contentious, brothel operators set the price and percentage cut, and the lack of contracts made conditions of work unclear.

Employment rights are not easily enforceable or gained while operating under an often-vague status as independent contractors (Gall 2016; Murray 2001). Only an employee can take a grievance to the Employment Court in New Zealand. However, a 2004 amendment to the Employment Relations Act 2000 made provision for those not in an employment relationship to take disputes to the Department of Labour and obtain free mediation services (Prostitution Law Review Committee 2008). Alternatively, sex workers who experience a breach of human rights in the workplace can take the matter to the Human Rights Tribunal. Some sex workers in this study reported willingness to seek recourse when denied their rights, either through interaction with the police, or through such formal complaint mechanisms. However, frequently, the first port of call is NZPC, who play a mediating role between brothel operators and sex workers, and issues have been resolved in many instances to everyone's satisfaction. There is a cost to pursuing complaints further up the chain than NZPC. One of the costs is the possibility of having their sex work status exposed. Sex workers see this as a very real risk in spite of the fact that to-date, all sex workers who have reported incidents have had name suppression. Sex work still carries a stigma in a decriminalised context. Additionally, as independent contractors, sex workers are required to pay taxes and contribute to the Accident Compensation Corporation (ACC). Some sex workers do declare their earnings, but many do not. Some sex workers also top up their income from sex work by claiming unemployment benefits. Putting their head above the parapet by filing an official complaint against an operator for unfair practices therefore carries a significant risk. They see it as simply more expedient to move to work in another brothel.

## 5. Conclusions

This study contributes to a body of research that indicates that decriminalisation increases the likelihood of commercial sex being practised consensually and in a more worker-controlled manner and therefore can be considered a prerequisite for the realisation of sex workers' rights (Pitcher and Wijers 2014). The brothel environment in New Zealand today is vastly different from that which Plumridge (2001) discussed in her late 1990s study. Exploitative practices were commonplace in brothels, and sex workers could do little about it. Sex work remains precarious work in New Zealand. Sex workers receive no base fee and no guarantee that they will get clients on their shifts, and they are vulnerable to dismissal without warning. However, decriminalisation has contributed to an important recognition of sex workers' rights and provided them with protections in the workplace. It is now possible for sex workers to experience safer and more supportive work environments than they otherwise might, where they can (and sometimes do) contest managerial control. The addition of the right to refuse to provide commercial sexual services in the Prostitution

Reform Act has been particularly crucial to changing management practices in brothels and had a meaningful impact on the way sex workers in this study negotiated potentially exploitative dimensions of brothel-based work. They were prepared to challenge brothel operators who impinged on their rights, which meant that often operators backed down. They were prepared to entertain the idea of laying official complaints against offending operators—and some did. While the majority might decide against such measures, they at least do have the opportunity. This opportunity is less viable when sex work is criminalised. There may still be some exploitation in New Zealand brothels, but the power of brothel operators has diminished considerably with the recognition of sex workers' rights. Emphasis in the future should be on encouraging more sex workers to challenge their working conditions, as well as the stigma and discrimination they experience because of their occupational choice.

**Author Contributions:** G.A. conceived and designed the study, undertook the analysis and interpretation of the findings, and produced the final draft of the paper. M.L. produced the first draft of the paper, and edited and approved the final version. All authors have read and agreed to the published version of the manuscript.

**Funding:** The University of sOtago Medical Research Foundation provided the funds needed to support this research.

**Institutional Review Board Statement:** The study was conducted according to the guidelines of the Declaration of Helsinki, and approved by the the University of Otago Human Ethics Committee (Ref no: 15/169 on 15 December 2015).

**Informed Consent Statement:** Informed consent was obtained from all subjects involved in the study.

**Data Availability Statement:** The data presented in this study are available on request from the corresponding author. The data are not publicly available due to confidentiality agreements with participants.

**Acknowledgments:** We acknowledge our research partners, New Zealand Prostitutes' Collective, and in particular Dame Catherine Healy who has provided valuable input to this paper.

**Conflicts of Interest:** The authors declare no conflict of interest.

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
