# Peer review of "Brothels as Sites of Third-Party Exploitation? Decriminalisation and Sex Workers’ Employment Rights"

_socsci, doi:10.3390/socsci10010003_

Round 1

Reviewer 1 Report

This is an important addition to the literature on working conditions of sex workers under Decriminalisation in New Zealand. It is encouraging to see the focus in this paper moving beyond the usual criminal justice debates here to consider sex worker's labour conditions, rights and working relationships.  Gives an idea of what might be more fruitfully focused on under more enlightened governance. Important to see the voices and experiences of sex workers centrally placed in the article. This is an important and welcome addition to the literature.

One comment:

If there is space, I was interested to learn more about the use of CPAR in this study? I'm not entirely clear on the approach used. Was the study also co-designed with sex workers? Is CPAR distinct from PAR? If so, why was CPAR chosen over a more standard PAR approach? 

Author Response

I have addressed the reviewer's comment  in lines 113-117 by expanding on the difference between PAR and CBPR. The principles are very much the same between the two approaches.

Reviewer 2 Report

Dear authors, 

Thank you for submitting your manuscript. I enjoyed reading it. The article is well-written and offers an insightful contribution to the field of research on the decriminalization of sex work. Overall, I think that the article presents new and significant information and analytical insights related to the research topic that justifies its novelty. The theoretical framework is resumed in the introduction, where you discuss the relevant literature around the decriminalisation of sex work, and its impact on the experiences and agency of the sex workers. As regards the methodology, the research is well-designed, and the method employed was appropriated (interview) to the research aim. The results and discussion are well-articulated, and the “voices” of the interviewees are emphasised in the analytical work. Finally, the article is well written and is easily readable.

However, the article needs to develop and justify some few key points. I hope my few comments might be useful for helping the authors to develop further their work.

Literature review 

You demonstrate an adequate understanding of the relevant literature in the field and cite an appropriate range of literature sources. However, I think it would be useful to create a literature review section where they may explore in detail the extensive literature presented in the introduction and, by doing so, go further on the debate about the impact of decriminalization on the sex workers rights and management control;

Methodology (analysis procedures) 

In my opinion, you need to present in a mode detailed way how the thematic analysis was done and how the authors got to the three main themes: employment status, the economics of brothel-based work, and agency in a brothel setting. However, I wonder if were these themes the final codes of your process of coding. What about the first-order codes or second-order codes? 

Also, it would be useful to know how each author contributed to the analysis.

 Research contributions and implications to practice

The implications for theory need to be emphasised, since they are, somehow, diluted in the discussion section.

On the other hand, I advise the authors to go further on the implications for research and practice of their findings and to give concrete examples of how can these findings support institutional policies related to sex workers rights and the management practices of the brothel operators.

Author Response

I have addressed the reviewer's comments in the following ways:

Literature review - I acknowledge that there could have been a more detailed and extensive literature review but this would have pushed the word limit well over 10,000 words and I would have to shorten the findings section. I think that the literature review as it stands does set up the argument for the focus of the study though.

Methodology - I have provided more detail on thematic analysis and the coding in lines 152-159. "Thematic analysis was then undertaken, which is a method used extensively in qualitative research to identify, analyse and report patterns in data (Aronson, 1994; Braun and Clarke, 2006). Both authors read and re-read the transcripts to familiarize themselves with the data and then coded for features of specific significance. For this paper we were interested in the talk of both sex workers and brothel operators on management practices. Codes included (amongst many others) the law, contractors, employees, shift fees and fines, resistance, co-dependency and reporting.  Codes were then collapsed into overarching themes." 

The breakdown of each author's contribution to the manuscript is provided in the "Authors' contribution" section at the end of the paper. 

Implications - Some additions have been made in the conclusion, lines 555-572, on implications.